# The Effects of Sterilization Procedures on the Cutting Efficiency of Endodontic Instruments: A Systematic Review and Network Meta-Analysis

**DOI:** 10.3390/ma14061559

**Published:** 2021-03-22

**Authors:** Mario Dioguardi, Enrica Laneve, Michele Di Cosola, Angela Pia Cazzolla, Diego Sovereto, Riccardo Aiuto, Luigi Laino, Teresa Leanza, Mario Alovisi, Giuseppe Troiano, Lorenzo Lo Muzio

**Affiliations:** 1Department of Clinical and Experimental Medicine, University of Foggia, Via Rovelli 50, 71122 Foggia, Italy; enrica.laneve@unifg.it (E.L.); dott.dicosola@gmail.com (M.D.C.); elicio@inwind.it (A.P.C.); diego_sovereto.546709@unifg.it (D.S.); giuseppe.troiano@unifg.it (G.T.); lorenzo.lomuzio@unifg.it (L.L.M.); 2Department of Biomedical, Surgical, and Dental Science, University of Milan, 20122 Milan, Italy; Riccardo.Aiuto@unimi.it; 3Multidisciplinary Department of Medical-Surgical and Odontostomatological Specialties, University of Campania “Luigi Vanvitelli”, 80121 Naples, Italy; luigi.laino@unicampania.it; 4Azienda Ospedaliera Universitaria, University of Campania “Luigi Vanvitelli”, 80121 Naples, Italy; teresa.leanza@policliniconapoli.it; 5Department of Surgical Sciences, Dental School, University of Turin, 10127 Turin, Italy; mario.alovisi@unito.it

**Keywords:** endodontic, cutting efficiency, mechanical property, network meta-analysis, autoclave, sterilization

## Abstract

Sterilization processes guarantee the sterility of dental instruments but can negatively affect instrument features by altering their physical and mechanical properties. The endodontic instrumentation can undergo a series of alterations, ranging from corrosion to variation in the cutting angle and then changes in the torsional properties and torsional fatigue resistance. This systematic literature review and meta-analysis aims to investigate alterations to the cutting efficiency of endodontic instruments that are induced by procedures for their disinfection and sterilization. Methodologies adopted for this investigation follow the PRISMA (Preferred Reporting Items for Systematic Reviews and Meta-Analysis) guidelines. The following were used as search terms on PubMed and Scopus: “endodontic sterilization”, “endodontic autoclave”, “cyclic fatigue”, “torsional”, “cutting efficiency”, “sterilization”, “surface characteristics”, and “corrosion”. At the end of the selection process, 36 articles were identified, and seven of them are included in this systematic review. The results of a meta-analysis conducted for the use of 10 autoclaving cycles shows a standardized mean difference (SMD) of 0.80 with a *p*-value equal to 0.04 with respect to effect on cutting efficiency. The network meta-analysis, through direct and indirect comparison between the different autoclave cycles (0, 1, 5, 10, and 15 cycles), revealed that treatment involving 15 autoclave cycles produced the most robust results in terms of having the greatest effects in terms of altered cutting efficiency with a probability of 57.7% and a SUCRA (surface under the cumulative ranking) of 80%. The alterations in the effects on cutting efficiency appear to be triggered after five cycles of sterilization by heat (autoclave). In conclusion, the meta-analysis of the data indicates that the autoclave sterilization protocol must not be repeated more than five times to preserve cutting efficiency. Within the limitations of this review, we can therefore establish that sterilization by autoclaving alone results in steel and NiTi instruments becoming less efficient in cutting after five cycles, as measured by a reduction in cutting efficiency.

## 1. Introduction

The sterilization of endodontic instruments represents an important process for proceeding with the reuse of endodontic instruments. Sterilization has the purpose of breaking down and eliminating all microorganisms, viruses, and spores and preventing cross-infections [1].

The sterilization process includes several phases: pre-sterilization, drying, packaging, heat sterilization, and storage of the sterile material [2].

Pre-sterilization consists in the disinfection, decontamination, and cleaning of the dental instruments. This phase involves disinfection either by immersion of the instruments with decontaminating and disinfecting liquids or by washing with vanish disinfectors. The purpose of decontamination is to achieve a reduction in the microbial load, while cleansing aims to remove organic and inorganic residues from endodontic instruments (to avoid the removal of debris through hand-brushing, the use of ultrasonic trays is recommended) [3].

The subsequent phases involve the rinsing, drying, and packaging of instruments followed by heat sterilization (autoclave at 134 °C at 2 bar) to eliminate spores. The last phase is storage of the instruments [4].

The fracture of endodontic instruments inside the canal is a problem that is not always easy to resolve [5]. The possibility of reusing many endodontic instruments after sterilization procedures raises the question of how much these procedures can influence the physical and mechanical properties, with three possible answers: a worsening, an improvement, or no effect [6].

The literature is not in agreement that there is an absolute lack of influence of procedures on instrument properties. Zhao et al. (2016) reports, as regards HyFlex CM, Twisted File, and K3XF instruments (autoclave sterilization performed at 134 °C with a pressure of 30 psi for 5 min), an increase in resistance to cyclic fatigue [7]. Viana et al. (2006) reports an average of cycle numbers (916–950 cycles to failure) associated with higher failure in heat-sterilized profiles [8]. These are in contrast with recent study conducted by Masoud Khabiri et al. (2017) on NiTi instruments, which reported no influence on cyclic fatigue [9], while Silvaggio and Hicks (1997) demonstrate that 10 autoclave cycles do not increase the risk of fracture in profiles [10]. Resistance to cyclic fatigue is not the only physical and mechanical property affected by sterilization procedures—e.g., there is also the resistance to torsional fatigue, about which recent studies disagree with respect to improvement, as stated by Casper et al. using M Wire alloys, and the deterioration, as reported by King et al. using the Gt x series [11,12]. Furthermore, sterilization procedures can negatively affect the cutting efficiency and create corrosive effects on the surface of instruments, as demonstrated in a 2018 study by Nashwan-Ahmed Qaed et al., which reports a reduction in the cutting angle and surface modifications using Revo S instruments subjected to autoclaving [13].

Scientific studies therefore report conflicting opinions on the alterations affecting endodontic instruments subject to sterilization procedures. This divergence of opinions and results could depend on the countless instruments and protocols used in endodontics and on the different sterilization and disinfection methods [1].

Previous revisions on the subject of the same research group have concerned aspects of the torsional property and surface alterations of the endodontic instruments subject to sterilization, focusing neither on cyclic fatigue nor on cutting efficiency [4,14,15]. There are currently no systematic reviews focusing on alterations in cutting efficiency in relation to sterilization.

The purpose of this review is to provide the broadest and most updated overview of the topic using systematic methodology for the review process [16,17]. Our hypothesis is that there is a reduction in the cutting efficiency on rotary endodontic instruments induced by autoclaves regardless of clinical use.

## 2. Materials and Methods

The following review was performed in accordance with the indications of PRISMA (Preferred Reporting Items for Systematic Reviews and Meta-Analysis) [17]. The methodology used for the drafting of the systematic review was adopted by previous systemic reviews of the same research group in the same research area [14,15]. PICO is defined as follows:

Participants: endodontic instruments.

Intervention: sterilization cycles through the use of an autoclave.

Comparison: endodontic instruments that have not undergone heat sterilization processes.

Outcome: reduction in the cutting capacity of endodontic instruments subjected to an autoclave cycle, expressed as the difference between the mean.

The primary outcome of this systematic review is therefore to answer the following question: “To what extent can the sterilization processes modify or reduce the cutting efficiency of endodontic instruments?”.

After an initial screening phase, eligible articles were studied for qualitative and quantitative analysis to investigate the influence of the sterilization procedures on the cutting efficiency of endodontic instruments used for the scouting, glide path, and shaping of the endodontic canal.

### 2.1. Eligibility Criteria

The included studies were in vitro studies concerning the subject of sterilization and, in particular, the influence of disinfection and sterilization procedures on endodontic instruments that were conducted within the last 40 years and published in English. Articles from within the last 40 years were chosen because disinfection and sterilization procedures have changed radically in light of newly discovered infectious contaminants, such as the HIV and HCV viruses and the prion of spongiform encephalopathy [18]. Furthermore, the methods used to manufacture the instruments have also changed, with the introduction of new alloys and new instruments in recent decades [19,20].

### 2.2. Research Methodology

Studies were identified through bibliographic searching of electronic databases.

The literature search was conducted using the search engines “PubMed” and “Scopus”. The search was conducted between 11 January 2019 and 14 February 2019, and the final search for a partial update of the literature was conducted on 19 September 2020.

The following search terms were used on PubMed and Scopus: “Endodontic sterilization” “endodontic autoclave”, “cyclic fatigue” AND “sterilization”, “torsional” AND “sterilization”, “cutting efficiency” AND “sterilization”, “surface characteristics” AND “sterilization”, corrosion” AND “sterilization” (Table 1).

### 2.3. Screening Methodology

Record screening was conducted by two independent reviewers, and any disagreements were resolved by a third reviewer. The screening included the analysis of the title and abstract to eliminate records not related to the themes of the review. Overlaps were then eliminated. The articles considered potentially eligible were studies on the influence of sterilization and disinfection procedures regarding the physical and mechanical characteristics of endodontic instruments. The potentially eligible articles were finally subjected to a full text analysis to verify their use for qualitative and quantitative analysis; any disagreements were resolved by a third reviewer.

The inclusion criteria applied to the qualitative analysis include all studies that discussed methods for the sterilization of endodontic instruments. The exclusion criteria implied the exclusion of all studies that did not discuss the influence of sterilization methods on the cutting efficiency of endodontic instruments. Studies were excluded if they did not report data on cutting efficiency or depth of cut, were not written in English, or were published before 1980.

Two reviewers performed the research and screening of the articles: M.D., DDS, and first-year post-doctorate, and E.L., post-graduate. The third reviewer was G.T., DDS. These three reviewers are from the Department of Clinical and Experimental Medicine of the University of Foggia (Foggia, Italy). The fourth reviewer, with supervisory duties, was L.Lo.M., DDS MD.

### 2.4. Statistical Analysis Protocol

The protocol used for the meta-analysis was based on the written indications from the Cochrane Handbook for the systematic review of interventions. The software used for data analysis was Rev Manager 5.4 (Cochrane Collaboration, Copenhagen, Denmark); in particular, the average difference (of the cutting depth) in the 2 groups (no autoclave, autoclave) divided by the cumulative estimate of the standard deviations was measured, and the presence of heterogeneity was measured w\ith the Higgins index (*I*^2^); values greater than 50% were considered significant. The heterogeneity of the studies was also evaluated graphically using funnel plots. A sensitivity analysis was also conducted through testing the exclusion or inclusion of data from the included studies. Finally, the Stata 14 (StataCorp LP, College Station, TX, USA) program was used for the execution of the network meta-analysis, having the various protocols of autoclave cycles as different treatments.

## 3. Results

A total of 894 records were identified on the PubMed and Scopus databases. The search results for each individual keyword are as follows: “Endodontic sterilization”: PubMed: 316; Scopus: 236; “endodontic autoclave”: PubMed: 37; Scopus: 50; “cyclic fatigue” AND “sterilization”: PubMed: 19; “torsional” AND “sterilization”: PubMed: 30; “cutting efficiency” AND “sterilization”: PubMed: 13; “surface characteristics” AND “sterilization”: PubMed: 29; “Corrosion” AND “sterilization”: PubMed: 137 (Table 1).

After proceeding with the screening of articles with restriction according to the year of publication (last 40 years), there were 832 records. After application of the eligibility criteria (all articles about sterilization in endodontics), we had 161 articles, and after the elimination of 57 article overlaps (the overlaps were removed using EndNote 9 software) and with the application of the exclusion and inclusion criteria, seven articles investigating the cutting efficiency (evaluated in the qualitative analysis) were obtained, and three articles were included in the meta-analysis.

The whole selection and screening procedure, as described in Table 1, is represented in the flow chart (Figure 1).

### 3.1. Study Characteristics and Data Extraction

The studies included for the qualitative analysis were those of Schafer et al. (2002) [21], Haikel et al. (1996) [22], Rapisarda et al. (1999) [23], Seago et al. (2015) [24], Haikel et al. (1998) [25], Neal et al. 1983 [26], and Morrison et al. (1989) [27].

The following three studies were included in the meta-analysis: Schafer et al. (2002) [21], Neal et al. 1983 [26], and Morrison et al. (1989) [27].

Data extraction and the methods of reporting were performed according to the Cochrane Handbook for Systematic Reviews of Interventions, Chapter 7 (Selection of Studies and Data Collection), specifically from pages 152 to 182. The extracted data included the journal (author and data); type of endodontic file investigated and the number of instruments; the type of autoclave adopted with the relative sterilization cycle use, temperature, and time; investigation methods to detect alterations and the main results of the study. The data extracted are shown in Table 2.

### 3.2. Risk of Bias

The risk of bias was assessed based on the CRIS (Checklist for Reporting In Vitro Studies) Guidelines [28], proposed to evaluate in vitro dental studies. Results are shown in Table 3. Each category was assigned a value from 1 to 5 (where 1 = low and 5 = high). The questions that the review answered, by assigning a score, were the following:Sample size calculation: Is the sample size adequate for obtaining statistically significant results?Meaningful difference between groups: Has the “meaningful difference” measurement been set correctly in the groups taking into account the sample size and the type of measurement?Sample preparation and handling: Does the study describe information on the production or handling of the samples to be tested?Allocation sequence, randomization, and blinding: Did the samples have equal and independent possibility of a sample entering any group?Statistical analysis: Are the statistical methods described?

Studies presenting a high risk of bias were not included in the meta-analysis. Items with a high risk of bias were excluded from the scale and eliminated during the inclusion phase. Other articles were excluded because they presented the same data and samples for the investigated results. The assessment of the risk of bias of the three included articles was conducted by the first reviewer (M.D.).

The risk of bias among studies is considered high. Heterogeneity showed a value represented by *I*^2^ (Higgins index) of 65% with a *p*-value < 0.04 based on the chi-squared test (heterogeneity values greater than 50% are considered high). The high heterogeneity is also confirmed by the funnel plot in Figure 2. Schafer et al. (2002) [21] reported data on two different types of instruments (NiTi Kfile, NiTi Kfile PVD). It was therefore decided both that data in the meta-analysis would be used and that any removal of the data to study the variation on the final effect would be deferred to the sensitivity analysis.

### 3.3. Meta-Analysis Results

The statistical analysis of the data was performed using Rev Manager 5.4 (Cochrane Collaboration, Copenhagen, Denmark). The results are represented by forest plots in Figure 3.

The meta-analysis was conducted by applying random-effects models given the high rate of heterogeneity (65% with a *p*-value = 0.04) and by assessing the standardized mean difference (SMD) (the difference in the means of the groups was divided by the cumulative estimate of the standard deviation). The meta-analysis showed an SMD value of 0.80 CI (confidence interval) [0.05, 1.55] (*p*-value = 0.04), revealing a significant association between autoclaving and reduction in cutting depth.

In addition, a sensitivity analysis was performed, aimed at verifying the robustness of the meta-analysis data by alternately excluding the data provided by Schafer et al. (2002) [21]. After the exclusion of this study from the meta-analysis, a significant difference between the test and control groups was obtained. In particular, the outcome was in favor of the control group (0 cycles), in the first case, with a reduction of the SMD to 0.45, *p*-value = 0.06, and in the second, with an increase of SMD to 0.99, *p*-value = 0.06, *I*^2^ = 72% (Figure 4).

The meta-analysis compared the data of the studies concerning only two different groups (instruments that have undergone 10 cycles of autoclaving and instruments have not undergone sterilization processes). We performed the network meta-analysis in order to compare different sterilization treatments by autoclaving (0, 1, 5, 10, and 15 cycles) using indirect and direct comparisons of the various treatment protocols. Data included in the network meta-analysis were related to the principle of similarity since the methodology adopted is similar in each study making them therefore comparable. The network plot shown in Figure 5 shows how the main data of the direct comparisons are mainly on Nodes 0, 5, and 10, with robustness in the lines of the comparisons between Nodes 10–5, 10–0, and 5–0.

The general consistency test reports a Chi^2^ (2) = 0.88, *p* = 0.645. Table 4 instead shows the results of the local incoherence test, listing the extent of the differences for each treatment and the results of the statistical test. None of the treatments showed statistical significance.

Since the inconsistency was absent in both global and local tests, the incoherence hypothesis was accepted.

For the analysis of the effect size, we proceeded through the representation by forest plot (Figure 6).

It should be noted that the *p*-value displayed in the lower left part of the graph is congruent with the results of the global test on inconsistency, which confirms that the coherence is accepted.

The values of heterogeneity between individual studies within a treatment are confirmed. In addition, based on the similarity between the aggregate effect size of each treatment in the comparison set (blue color) and the aggregate overall effect size (red color), it can be confirmed that the consistency model is supported.

Having assessed the comparative effectiveness of the treatments through the previous steps, we ranked the treatments to identify superiority. We then evaluated the treatment interventions that show the highest treatment effect (Table 5). As shown in Figure 7, the probability that Treatment 15 (15 cycles) is the best (resulting in a greater reduction in cutting efficiency) is approximately 57.7%, and the probability that it is at least second best is 79.3%. In fact, in the SUCRA (surface under the cumulative ranking), the surface for the treatment of 15 autoclave cycles reaches almost 80%, confirming once again that it is the treatment that determines the greatest alteration.

## 4. Discussion

### 4.1. Sterilization Methods

The disinfection and sterilization procedures of the endodontic instruments can vary depending on the material, whether they are single-use or reusable (through hot sterilization procedures, which require the use of the autoclave) [29].

The first-use instruments must necessarily undergo a cleansing and decontamination phase to remove metal residues (chromium nickel residues present on the surface deriving from the production processes) [30]. In fact, Filho et al. stated the need to remove metal residues from endodontic files prior to clinical use or sterilization [31]. Hauptman et al. also demonstrated the presence of microorganisms (*P. lentimorbus*), even in the presence of instruments supplied as sterile by the manufacturing industries. The presence of these microorganisms underlines how important it is to decontaminate the instruments before use, which do not, however, require sterilization by heat that may alter their physical, mechanical, and surface properties [32].

These operations can be performed with the use of an ultrasonic bath or a disinfected tray and with a few minutes of immersion in disinfectant solutions such as 2% chlorhexidine or 2% peracetic acid [33].

These first-use or non-autoclavable instruments, after being rinsed with distilled water, are dried, packaged, and ready for clinical use [1].

The reusable instruments used for the various operational phases, such as scouting glide path and channel shaping, present critical issues (not only related to the influence of the mechanical and physical properties), such as the risk of cross-infection for health workers or patients, and the risk related to the presence of nonviral and nonbacterial contaminants, such as prions of human spongiform encephalopathy [34]. We recall that prions are only partially inactivated by normal sterilization procedures (autoclave). In fact, it is known that the manual removal of residues between the blades is ineffective without ultrasound. Moreover, the hand-brushing procedure presents a greater risk of cross-infection for decontamination and disinfection operators [35].

Heat sterilization can be performed using 3 different methods:Steam sterilization (autoclaves): The dried and packaged instruments undergo a heating cycle (transmitted through the steam that penetrates the packaging) that can vary from 134 °C to 121 °C for a period of time that can vary from 5 min to 30 min [36,37,38].

Dry heat sterilization: The heat is not transmitted through the steam but through the air that can be forced to circulate at a temperature of 190 °C for a period of time ranging from 5 min to 15 min up to 2–4 h, dependent on the packaging method (metallic packages or packages out of plastic laminates that can withstand much heat) [39].

Sterilization by chemical steam (chemiclaves): A sterilizing agent is used, and a chemical steam (alcohol and formaldehyde) is suitably heated. The sterilization cycle is 20 min at 134 °C [37,38].

Most of the instruments are autoclavable at a temperature of both 120 °C and 135 °C but must first undergo a decontamination phase for the removal of residues and microorganisms (ultrasound trays or vanish disinfectors with the combined use of detergents and disinfectants) [40], followed by washing with distilled water, drying, and packaging [1].

Endodontic instruments can be influenced by sterilization and disinfection processes at various times. The analysis of the literature shows how these alterations can come from two phases:The disinfection phase, by physical chemical action by decontaminating agents [41].The sterilization phase, by autoclave for temperature action [42].

These two phases determine the major influences on metal endodontic instruments.

The effects can be summarized as follows:Corrosive effects both from disinfectant agents (sodium hypochlorite) with the phenomenon of micropitting, and from oxygen, with the formation of NiTi oxides under thermal stress in an autoclave [43].An increase in surface roughness of the surface in nickel titanium following autoclave sterilization [44].Partial recovery of macroscopic deformities in NiTi instruments after sterilization treatment in the autoclave [45].Partial recovery of cyclic fatigue suffered by NiTi instruments (most but not all studies) in the autoclave [46].Partial recovery of the torsional stress of the NiTi instruments, but not the totality of the studies, in the autoclave [47].Reduction in the cutting angle and of the resistance of endodontic instruments in steel subjected to the autoclave and partially for NiTi [21].Reduction in cutting efficiency (untreated NiTi alloys during the manufacturing process with hot procedures) [48,49].

### 4.2. Effect of Sterilization on Cutting Efficiency

One of the effects of sterilization procedures is the reduction in the cutting efficiency of the instruments. A study by Schafer et al. showed that there is a reduction in cutting efficiency in the NiTi K files equal to 16.1% after five sterilization cycles and 50.8% after 10 sterilization cycles by autoclave at a temperature of 135°. The same study draws attention to the fact that treatment with sodium hypochlorite at a concentration equal to 5.25% for 30 min has no effect on the reduction of cutting efficiency [21].

The data reported in the studies by Haikel et al. in 1996 and 1998 are similar. In this case, for sodium hypochlorite applied from 12 h to 48 h at 2.5% and ultrasonic cleaning baths, a reduction in cutting efficiency was reported for both single rows and flex on the file, with percentages ranging from 20% to 60% [22,25]. A minimal reduction in the cutting efficiency was also noted for glass bead sterilization and for dry hot sterilization for cycles of 5 or 10 min [25]. In this study, unifiles are endodontic files that have less reduction in cutting efficiency after sterilization procedures.

Morrisoon et al. tested K steel files following the use of extracted teeth and autoclave sterilization. The authors came to the conclusion that there is no reduction in the depth of the cut following five autoclave cycles and that the reduction is due only to clinical use [27].

### 4.3. Materials

#### 4.3.1. NiTi

Rapisarda et al. reported that there is a reduction in cutting efficiency even for rotary NiTi tools used for shaping, such as profiles. In their study, the reduction in cutting efficiency was 20% for seven cycles and 50% for 14 cycles (hot sterilization). Rapisarda et al. indicate the formation of NiTi oxides on the surface as a mechanism for the loss of cutting efficiency. Therefore, according to the authors, the phenomenon must be related to the corrosion induced by the oxidation of the surface layers [23].

Seago et al., in apparent contrast with the previous studies, do not notice any statistically significant reduction of the cutting efficiency on Hflex (NiTi instruments) for sterilization procedures by autoclave (1–2 cycles) [24].

The effects of steam sterilization in the investigated studies would appear after 5 sterilization cycles and become evident after 10 [22]. Moreover, the steel instruments and those in NiTi, such as the first-generation profiles, would be more subject to this effect. Treatment with sodium hypochlorite at concentrations ranging from 2.5% to 5.25% for 30 min did not show any effect on cutting efficiency. However, if the time reached 12–48 h, corrosive phenomena are triggered with a clear reduction in cutting efficiency [50]. On the NiTi instruments, the effects related to the reduction in cutting efficiency were found to be due to the formation of a layer of NiTi oxides on the instrument surface. The studies concerning the reduction in cutting efficiency that show apparent disagreement are listed in Table 2.

#### 4.3.2. Stainless Steel

The effects autoclaving on the mechanical properties of stainless steel instruments have been known for a long time. Mitchel et al. reported, in 1982, a significant reduction in the resistance to torque, with a reduction of the deflection angle during torsional stress after 10 autoclave cycles [51], data partially confirmed by Haïkel et al. [52]. Hilt et al. reported data indicating that neither the number of sterilization cycles nor the type of autoclave unit affect the torque, hardness, or microstructure of stainless steel in a statistically significant manner [53].

There are three studies in the literature on the cutting efficiency of stainless steel endodontic instruments as affected by autoclaving that were included in this review: Morrison et al. [27] reported no statistically significant difference for Flexofile instruments from 1 to 15 sterilization cycles, Neal et al. [26] described a small but significant decrease in the cutting capacity of files (K-type 30 #) after 10 autoclaving cycles, and Haïkel et al. found, using the Unifile and Flexofile, a cutting efficiency reduction (range of 20% to 70%) from five to 10 autoclave cycles [22].

### 4.4. Investigation Methods

Haikel et al. [54] described different methods used for measuring cutting effectiveness. The techniques used can be classified into two groups based on whether the tested instruments underwent a linear [55] or rotary movement along its long axis [54,56]; moreover, the simulated samples can be in plexiglass, dentin [57], or bovine bone [58].

The first study on the cutting efficiency of endodontic instruments was conducted by Molven et al. Dentin from sections of the extracted premolars was used as a study sample, and the instruments were used with mechanical and manual push and pull movements (linear motion). The created sulcus surface was measured without checking the load on the instrument [57].

Prior to this, Fromme and Riedel qualitatively assessed the cutting efficiency of reamers and H files and subject them to evaluations of images performed on tooth sections through scanning electron microscopy (SEM) [59].

Oliet was the first to introduce bovine dentine samples (with increased sample standardization) for the evaluation of cutting efficiency on reamers using a mechanical rotary motion with torque control without irrigation, and the depth of the grooves was measured at a millimeter scale [58].

Villalobos et al. investigated the cutting efficiency of the tip with rotary drilling movements on bovine bone maintained in humid conditions, with a measurement of efficiency through the measurement of time [56]. Webber et al. developed a system to measure the cutting efficiency in linear motion on blocks of bovine bone by testing different steel endodontic files with different section designs (a square reamer, a square file, a triangular file, a triangular reamer, and a Hedstroem file) [55].

In the studies included in the meta-analysis, the measuring devices involved measuring the depth of the cut. Morrison measured the depth of a cut on an apparatus (Quality Dental Products, Johnson City, TN, USA) that compares the sharpness of files when used in linear motion with irrigation, and blocks of phenolic resin were used as samples [27].

Neal measured the depth of the cut on methacrylate resin blocks with linear traction movement with irrigation, while Schafer et al. (2002) [21] measured cutting efficiency and the depth of the cut on plastic cylinders generated by the mechanical rotary movement of endodontic files [26].

SEM analysis highlighted the presence of topographical alterations on the surface. These alterations were visible already after the first cycle in the autoclave and can represent points where microcracks are triggered within the structure of the lega and determine cyclic fatigue failure or torsional stress [60,61].

The alterations induced on the surfaces, analyzed by SEM and AFM(Atomic Force Microscope), are also expressed on the cutting surface, detrimentally altering the cutting efficiency.

Surface analysis conducted by Inan et al., with the AFM reporting statistically significant data for all the instruments of the ProTaper series, noted the greatest effects on the ProTaper Finisching [62].

In 2012, Spagnuolo et al. confirmed that multiple sterilization cycles (autoclaving) changed the surface topography (examination conducted with AFM) and the chemical composition of conventional NiTi (F2 ProTaper) and TiN coated instruments (kit alpha) after five autoclave cycles [63].

Based on SEM analysis, Razavianet et al. reported an increase in roughness directly related to the number of sterilizations performed [64], while Nair et al. showed that, following sterilization, there is an increase in the surface roughness of the instruments due to an increase in the surface irregularities of the metal alloy, which could represent the cores from which the crack starts and the fracture of the tool occurs under cyclical fatigue [65].

### 4.5. Meta-Analysis

The results of the meta-analysis show that, by comparing the cutting efficiency of the instruments that performed 10 sterilization cycles with the control group, there is a statistically significant reduction in cutting efficiency, with an SMD of 0.80 CI [0.05, 1.55] and a *p*-value of 0.04.

The meta-analysis network directly and indirectly compared the various treatments (different sterilization cycles using an autoclave for 0, 1, 5, 10, and 15 cycles). The network meta-analysis highlights how the treatment that determines the greatest effect of reducing cutting efficiency is the 15 autoclave cycles, with a probability of 57% overall with a SUCRA equal to 80%.

The limitations of this meta-analysis are the small number of included studies that present a low risk of bias due to the different types of endodontic instruments investigated and the frequently varying methods for determining cutting efficiency.

We can establish that steam sterilization provides the best instrument sterility but will still negatively influence the cutting efficiency.

## 5. Conclusions

Our analysis of the scientific literature shows that there are various effects induced by the application of sterilization processes results in various effects on the physical and mechanical properties of endodontic instruments, some favorable and others unfavorable. Among the negative effects, the reduction in cutting capacity, that seems to emerge after five autoclave cycles (regardless of the clinical use), is fundamental, but not all studies reported similar data. Given the heterogeneity of the adopted measurement methods and the lack of in vitro studies, it is difficult to perform a meta-analysis of the data in the absence of bias between studies.

The effects on reduction in cutting efficiency are reflected in the majority of the results of the studies included in the systematic review. A reduction in the cutting efficiency can lead to an endodontist exerting greater pressure on the instrument in order to maintain constant cutting capacity, which would lead to a greater risk of instrument fracture due to torsional stress.

## Figures and Tables

**Figure 1 materials-14-01559-f001:**
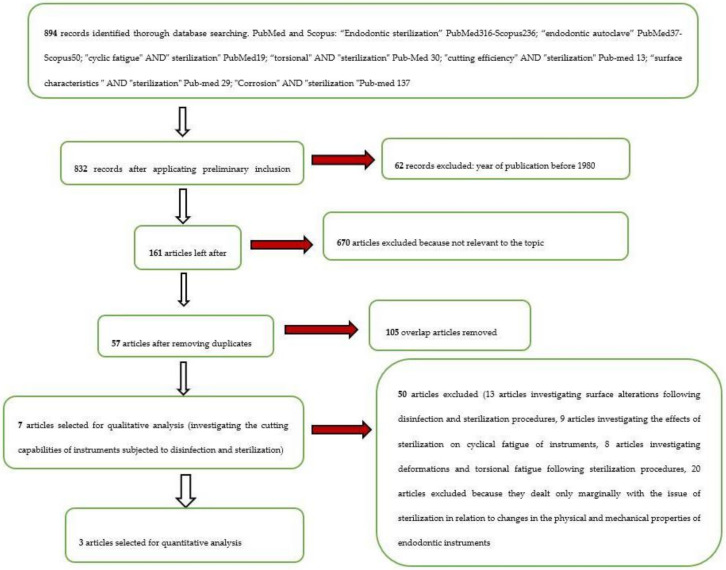
Flow chart of the different phases of the systematic review.

**Figure 2 materials-14-01559-f002:**
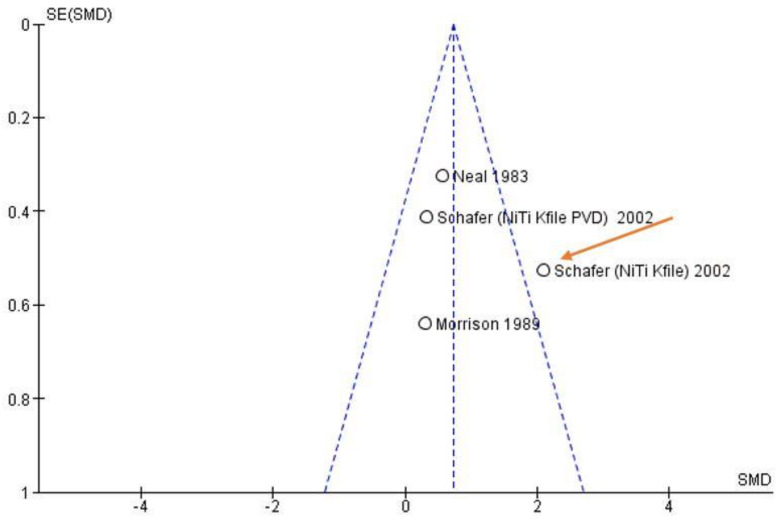
Funnel plot. The visual analysis highlights the heterogeneity between the studies (*I*^2^ = 65%). The source of the heterogeneity between the data of different studies is indicated with the orange arrow. The study, in fact, is placed outside the funnel. SE: standard error; SMD: standardized mean difference.

**Figure 3 materials-14-01559-f003:**
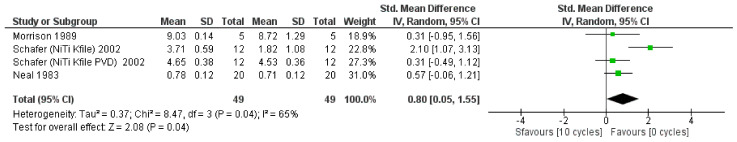
Forest plot showing a significant reduction in the cutting depth after 10 sterilization cycles by autoclaving compared to 0 cycles. The reduction in cutting efficiency was, on average, 0.80 μm higher in the test than in the control group. Legend: *Tau*^2^ = residual heterogeneity; Chi^2^ = chi-squared test; df = degrees of freedom; *I*^2^ = Higgins heterogeneity index; CI = confidence intervals; *p* = *p*-value; SD = standard deviation. The graph for each study shows the first author, date of publication, mean, standard deviation, and number of samples for the 2 groups (0 or 10 autoclave cycles). Furthermore, the standardized mean difference with associated confidence intervals and the weight expressed as a percentage of the total effect are reported.

**Figure 4 materials-14-01559-f004:**
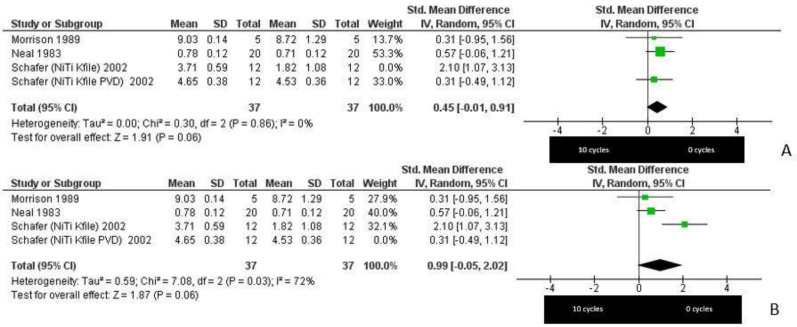
Sensitivity analysis. (**A**) excluded Schafer (NiTi Kfile) 2002: SMD = 0.45 CI [−0.01, 0.91], *p*-value = 0.06, *I*^2^ = 0%. (**B**) excluded Schafer (NiTi Kfile PVD) 2002: SMD = 0.99 CI [−0.05, 2.02], *p*-value = 0.06, *I*^2^ = 72%. A sensitivity analysis was performed by repeating the primary meta-analysis, including and excluding the data alternately to clarify whether the results are robust with respect to the decisions made in the review process. The sensitivity analysis, in this case, shows that the obtained results and conclusions were not influenced by the decision to include or exclude any studies; in fact, the final effect in favor of 0 cycles does not change. However, the heterogeneity of the study changed from *I*^2^ = 65% (Figure 3) to *I*^2^ = 72% (Figure 4B) and up to *I*^2^ = 0% (Figure 4A).

**Figure 5 materials-14-01559-f005:**
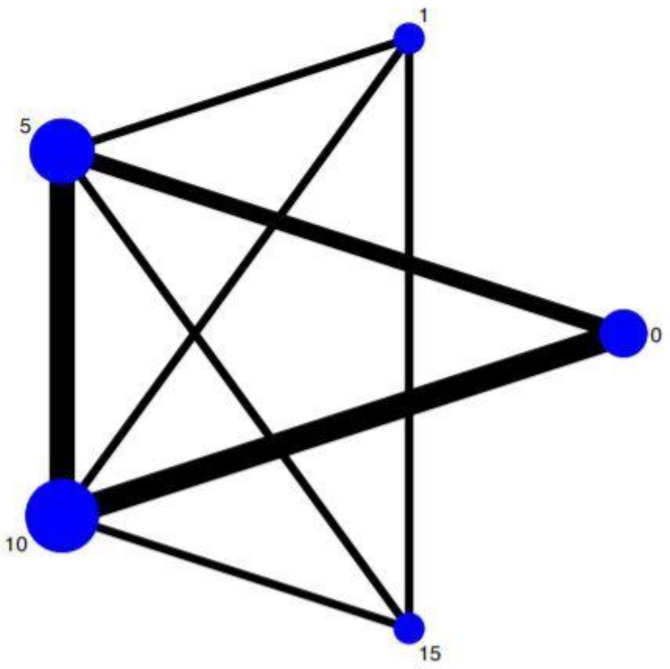
Network geometry. The size of the 5 nodes, one for each treatment, indicates the number of studies included in the corresponding nodes, while the thickness of the lines connecting 2 nodes indicates the amount of relevant data (0 (control), 1, 5, 10, and 15 cycles).

**Figure 6 materials-14-01559-f006:**
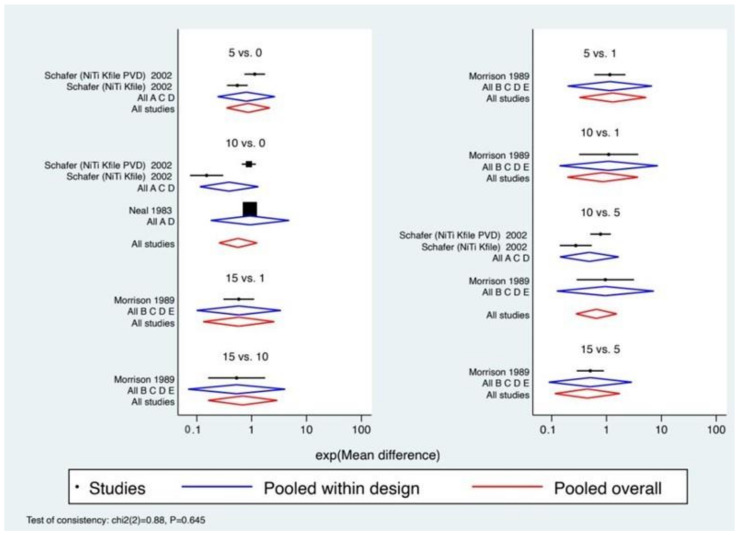
All direct and mixed comparisons. The pooled effect of a treatment in the comparison set (also called “pooled within design”) and the pooled overall effect (also called “pooled overall”) are marked in blue and red, respectively. The black squares in the center of the lines represent the estimate of the effect of a single study, and the black lines represent its confidence intervals. (A = 0 cycles (control), B = 1 cycles, C = 5 cycles, D = 10 cycles, and E = 15 cycles). The figure shows all the possible comparisons that emerged from the network analysis = 5 vs. 0 (Schafer -NiTi Kfile 2022, Schafer -NiTi Kfile PVD 2002), 5 vs. 1 (Morrison et al. 1989), 10 vs. 0 (Schafer -NiTi Kfile 2022, Schafer -NiTi Kfile PVD 2002, Neal et al. 1983), 10 vs. 1 (Morrison et al. 1989), 10 vs. 5 (Schafer -NiTi Kfile 2022, Schafer -NiTi Kfile PVD 2002, Morrison et al. 1989), 15 vs. 1 (Morrison et al. 1989), 15 vs. 10 (Morrison et al. 1989), and 15 vs. 5 (Morrison et al. 1989).

**Figure 7 materials-14-01559-f007:**
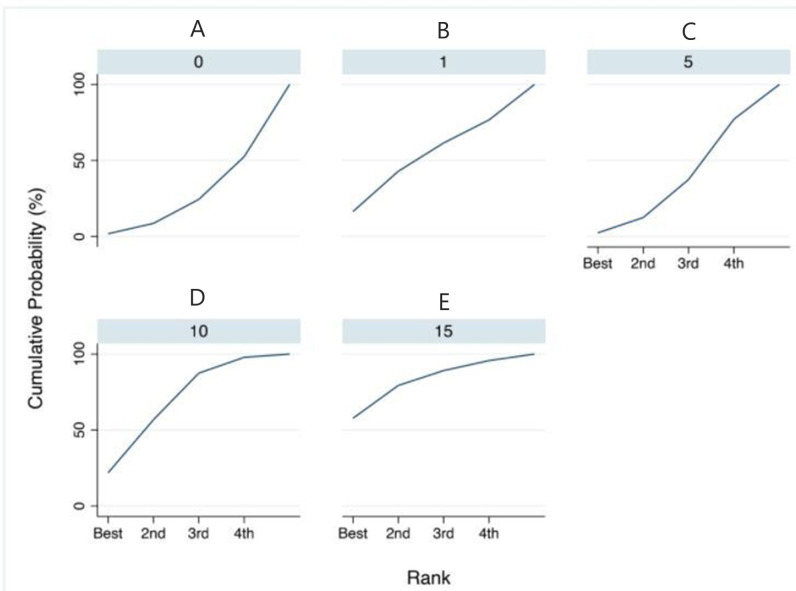
Results of the network rank test. Graphical ranking ((**A**) = 0 cycles (control), (**B**) = 1 cycles, (**C**) = 5 cycles, (**D**) = 10 cycles, (**E**) = 15 cycles); SUCRA: surface under the cumulative ranking. The higher the SUCRA value, the greater the probability that the cycles will cause an alteration in the cutting efficiency.

**Table 1 materials-14-01559-t001:** Overview of the search methodology; the total number of records was 161 after the initial screening and was 56 after removing overlaps.

Database—Provider	Keywords	Number of Records	Number of Records after Restriction by Year of Publication (Last 40 Years)	Number of Remaining Articles after Screening for the Latest Review Topic	Articles after Removing Overlaps	Articles Remaining after Applying the Inclusion and Exclusion Criteria	Articles Included in the Meta-Analysis
PubMed	“endodontic sterilization”	316	277	35	\	\	\
PubMed	“endodontic autoclave”	37	36	21	\	\	\
PubMed	“cyclic fatigue” AND “sterilization”	19	19	16	\	\	\
PubMed	“torsional” AND “sterilization”	30	30	17	\	\	\
PubMed	“cutting efficiency” AND “sterilization”	13	13	6	\	\	\
PubMed	“surface characteristics” AND “sterilization”	29	29	3	\	\	\
PubMed	“corrosion” AND “sterilization”	137	115	6	\	\	\
SCOPUS	“endodontic” AND “sterilization”	263	263	36	\	\	\
Scopus	“endodontic” AND “autoclave”	50	50	21	\	\	\
Total records		894	832	161	57	7	3

**Table 2 materials-14-01559-t002:** Characteristics of studies focusing on the cutting efficiency of endodontic instruments in relation to sterilization procedures.

First Author; Reference	Instruments	Autoclave Cycles	Number of Instruments	Depth of Cuts (mm), Mass of Cuts (µg)	Standard Deviation	Reduction of Cutting Efficiency, Expressed as a Percentage of the Control	Type of Autoclave	Time and Temperature	Results
Morrison et al. 1989 [27]	stainless steel #25 Flexofile	0	5	9.03	0.14		steam autoclave (Amsco Medalist 200, American Sterilizer Co., Erie, PA, USA)	15 min, 121 °C	no signilicant difference c in cutting efficiency
		1	5	8.63	0.58				
		5	5	8.77	0.47				
		10	5	8.72	1.29				
		15	5	8.09	0.45				
		1 after use 1 molar	10	8.34	0.77				
		5 after use 5 molar	10	7.07	0.83				
		10 after use 10 molar	10	6.84	0.69				
Schafer et al. 2002 [21]	NiTi Kfile 35#	0	12	3.71	0.59		Aesculap Automat 356 (Aesculap, Tuttlingen, Germany).	30 min, 134 °C	50.6–16.1% NiTi Kfile (10–5 cycles)
		5	12	3.11	0.49	16.1%			
		10	12	1.82	1.08	50.6%			
		5 + NaOCl treatment	12	3.06	0.65				
	NiTi Kfile PVD 35#	0	12	4.65	0.38				
		5	12	4.78	0.67				
		10	12	4.53	0.36				
		5 + NaOCl treatment	12	4.30	0.44				
Neal et al. 1983 [26]	K-type #30 stainless steel files	0	20	0.78	0.12		autoclave bags (Lorvic Corp., St. Louis, MO, USA)	30 min, 270 °F	autoclave sterilization resulted in a small but significant decrease in cutting ability of the files.
		10	20	0.71	0.12				
Haikel et al. 1996 [22]	stainless steel Unifile #30	0	10	132.4 ^1^ ± 0020			chemiclave	30 min, 131 °C	Unifile and Flexofile, a cutting efficiency reduction (range of 20% to 70%)
		5	10	38.2 ^1^ ± 0.0005		63.90%			
		10	10	35.9^1^ ± 0.0003		67.10%			
	stainless steel H-file #30	0	10	141.6 ^1^ ± 0.0020					
		5	10	43.1 ^1^ ± 0.0003		68.10%			
		10	10	46.8 ^1^ ± 0.0003		50.40%			
	stainless steel Flexofile #30	0	10	146.9 ^1^ ± 0.0030					
		5	10	34.7 ^1^ ± 0.0008		77.00%			
		10	10	39.7 ^1^ ± 0.0008		73.00%			
Rapisarda et al. 1999 [23]	NiTi ProFile instruments (#15, #30 #40 #45)	0	12				Euroclave (Euronda Spa)	30 min, 121 °C	the number of sterilization cycles was a determining factor as to cutting efficiency
		7	12			20% reduction in cutting ability			
		14	12			50% reduction in cutting ability			
Seago et al. 2015 [24]	NiTi Hyflex CM Rotary Files 35#	0, 1, 2, 3, 4, 5, 6, 7, 8, 9, 10 cycles	60				STATIM 5000 (SciCan, Toronto, ON, Canada)	6 min, 132 °C	no statistical decrease after 1, 4, 5, and 6 cycles. A statistically significant decrease in cutting efficiency was for 2 3 7 8 and 9 cycles
Haikel et al. 1998 [25]	NiTi file Maillefer #30	2.5% NaOCl, 12–48 h				(109.8% efficacy)–24.36%			
	NiTi file Brasseler #30	2.5% NaOCl, 12–48 h				16.42–32.87%			
	NiTi file JS Denta l#30	2.5% NaOCl, 12–48 h				29.97–25.35%			
	NiTi file McSpadden #30	2.5% NaOCl, 12–48 h				3.25%–no reduction			

^1^ The values refer to the cut mass. min = minutes; °C = centigrade; °F = Fahrenheit.

**Table 3 materials-14-01559-t003:** Assessment of the risk of bias within the studies, with scores 7 to 12 = low quality, 13 to 20 = intermediate quality, and 21 to 25 = high quality.

First Author; Reference	Sample Size Calculation	Meaningful Difference between Groups	Sample Preparation and Handling	Allocation Sequence, Randomization, and Blinding	Statistical Analysis	Score
Schafer et al. (2002) [21]	3	4	4	4	5	20
Neal et al. 1983 [26]	4	3	3	3	4	17
Morrison et al. (1989) [27]	2	4	3	3	4	16

**Table 4 materials-14-01559-t004:** Inconsistency test between direct and indirect treatment comparisons in a mixed treatment comparison (A = 0 cycles (control), B = 1 cycle, C = 5 cycles, D = 10 cycles, and E = 15 cycles).

Side	Direct		Indirect		Difference		
	Coef	Std. Err	Coef	Std. Err	Coef	Std. Err	*p* >|z|
A C	−0.2264099	0.5766333	0.1967006	1.256345	−0.4231105	1.381023	0.759
A D	−0.6247653	0.4634597	0.372054	2.195575	−0.9968193	2.249488	0.658
B C	0.1400003	0.8378915	1.136114	2.10521	−0.9961141	2.255608	0.659
B D	0.0900002	0.9953451	−0.9091564	1.889073	0.9991566	2.256189	0.658
B E	-	-	-	-	-	-	-
C D	−0.5406656	0.4789061	1.003904	1.739389	−1.544569	1.808625	0.393
C E	−0.6800003	0.8217556	−1.676114	2.09884	0.9961141	2.255609	0.659
D E	−0.6300001	0.09818004	0.3691565	1.881972	−0.9991566	2.256189	0.658

**Table 5 materials-14-01559-t005:** The columns show the probability expressed as a percentage that each treatment is the best (which determines the greatest effect on the cutting efficiency in terms of reduction).

	Treatment				
Sudy and Rank	0 cycles (control)	1 cycle	5 cycles	10 cycles	15 cycles
Morrison 1989					
Best	1.7	16.3	2.3	21.8	57.7
2nd	6.8	26.6	10.0	35.0	21.6
3rd	15.8	18.6	25.1	30.7	9.8
4th	28.1	15.2	39.7	10.4	6.6
Worst	47.5	23.3	22.8	2.1	4.3
Mean Rank	4.1	3.0	3.7	2.4	1.8
SUCRA	0.2	0.5	0.3	0.7	0.8

## Data Availability

Data sharing not applicable.

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
