# Peer review of "The Effects of Sterilization Procedures on the Cutting Efficiency of Endodontic Instruments: A Systematic Review and Network Meta-Analysis"

_materials, 2021, doi:10.3390/ma14061559_

Round 1

Reviewer 1 Report

The manuscript “The Effects of Sterilization Procedures on the Cutting Efficiency of Endodontic Instruments: Sistematic Review and network meta-analysis” must be revised seriously before publication. A lot of grammar mistakes are present and most of the figures from manuscript are unclear.

Also, the authors have some serious problems with the terms related to the materials (e.g. line 235-237 “3.2 risk of bias The Risk of Bias”, Stainless stell) and they are advised to check carefully the manuscript before resubmission.

The part “3. Results” must be correlated with the part “4. Discussion”. I suggest to use just 3.3. Metanalysis results instead of “3.3 metanalysis, 3.4 Network metanalysis, 4.3 meta-analysis results”.

Also, in part 4, the discussion must be related to the aspects “4.1. Cutting efficiency”, “4.2.Materials”, “4.3. Sterilization methods” and “4.4. Investigation methods”.

Anyway, the actual structure of the manuscript is not properly made.

Regarding the investigation methods, the authors must pay attention to the methods like scanning electron microscopy that reveal the surface deterioration and the mechanical properties that was modified due to the sterilization methods.

I suggest to the authors to check some references who describe some evaluation methods like SEM : Ni-Ti ROTARY INSTRUMENT FRACTURE ANALYSIS AFTER CLINICAL USE. STRUCTURE CHANGES IN USED INSTRUMENTS, T Stefanescu, IV Antoniac, RA Popovici, A Galuscan, T Tirca, Environmental Engineering & Management Journal (EEMJ) 15 (5); Adhesion aspects in biomaterials and medical devices, A Iulian, S Cosmin, A Aurora, Journal of adhesion science and Technology 30 (16), 1711-1715; Structural characterization and adhesion appraisal of TiN and TiCN coatings deposited by CAE-PVD technique on a new carbide composite cutting tool, AA Matei, I Pencea, SG Stanciu, R Hristu, I Antoniac, E Ciovica, CE Sfat, Journal of Adhesion Science and technology 29 (23), 2576-2589.

Author Response

Reviewer 1

The manuscript “The Effects of Sterilization Procedures on the Cutting Efficiency of Endodontic Instruments: Sistematic Review and network meta-analysis” must be revised seriously before publication. A lot of grammar mistakes are present and most of the figures from manuscript are unclear.

Also, the authors have some serious problems with the terms related to the materials (e.g. line 235-237 “3.2 risk of bias The Risk of Bias”, Stainless stell) and they are advised to check carefully the manuscript before resubmission.

The part “3. Results” must be correlated with the part “4. Discussion”. I suggest to use just 3.3. Metanalysis results instead of “3.3 metanalysis, 3.4 Network metanalysis, 4.3 meta-analysis results”.

Also, in part 4, the discussion must be related to the aspects “4.1. Cutting efficiency”, “4.2.Materials”, “4.3. Sterilization methods” and “4.4. Investigation methods”.

Anyway, the actual structure of the manuscript is not properly made.

Regarding the investigation methods, the authors must pay attention to the methods like scanning electron microscopy that reveal the surface deterioration and the mechanical properties that was modified due to the sterilization methods.

I suggest to the authors to check some references who describe some evaluation methods like SEM : Ni-Ti ROTARY INSTRUMENT FRACTURE ANALYSIS AFTER CLINICAL USE. STRUCTURE CHANGES IN USED INSTRUMENTS, T Stefanescu, IV Antoniac, RA Popovici, A Galuscan, T Tirca, Environmental Engineering & Management Journal (EEMJ) 15 (5); Adhesion aspects in biomaterials and medical devices, A Iulian, S Cosmin, A Aurora, Journal of adhesion science and Technology 30 (16), 1711-1715; Structural characterization and adhesion appraisal of TiN and TiCN coatings deposited by CAE-PVD technique on a new carbide composite cutting tool, AA Matei, I Pencea, SG Stanciu, R Hristu, I Antoniac, E Ciovica, CE Sfat, Journal of Adhesion Science and technology 29 (23), 2576-2589.

Answer

thanks for the comments and suggestions

  • As you suggested I have reviewed the manuscript and had it corrected at MDPI English Editing Services
  • (e.g. line 235-237 "3.2 risk of bias The Risk of Bias", Stainless stell) Checked and changed as required
  • The structure of the results and of the discussion has been changed, also accepting the suggestions of the reviewer 3
  • I have added a paragraph in the discussion on valuation methods taking into account the valuations performed to Sem and AFM, also including some articles suggested to me

SEM analysis highlighted the presence  of topographical alterations on the surface. These alterations were visible already after the first cycle in the autoclave and can represent points where microcracks are triggered within the structure of the lega and determine cyclic fatigue failure or torsional stress [1,2].

The alterations induced on the surfaces, analyzed by SEM and AFM, are also expressed on the cutting surface, detrimentally altering the cutting efficiency.

Surface analysis conducted by Inan et al., with the AFM reporting statistically significant data for all the instruments of the ProTaper series, noted the greatest effects on the ProTaper Finisching [3].

In 2012, Spagnuolo et al. confirmed that multiple sterilization cycles (autoclaving) changed the surface topography (examination conducted with AFM) and the chemical composition of conventional NiTi (F2 ProTaper) and TiN coated instruments (kit alpha) after five autoclave cycles [4].

Based on SEM analysis, Razavianet et al. reported an increase in roughness directly related to the number of sterilizations performed [5], while Nair et al. showed that, following sterilization, there is an increase in the surface roughness of the instruments due to an increase in the surface irregularities of the metal alloy, which could represent the cores from which the crack starts and the fracture of the tool occurs under cyclical fatigue [6]

Reviewer 2 Report

The manuscript “The Effects of Sterilization Procedures on the Cutting Efficiency of Endodontic Instruments: Sistematic Review and network meta-analysis” must be revised seriously before publication. A lot of grammar mistakes are present and most of the figures from manuscript are unclear.

Also, the authors have some serious problems with the terms related to the materials (e.g. line 235-237 “3.2 risk of bias The Risk of Bias”, Stainless stell) and they are advised to check carefully the manuscript before resubmission.

The part “3. Results” must be correlated with the part “4. Discussion”. I suggest to use just 3.3. Metanalysis results instead of “3.3 metanalysis, 3.4 Network metanalysis, 4.3 meta-analysis results”.

Also, in part 4, the discussion must be related to the aspects “4.1. Cutting efficiency”, “4.2.Materials”, “4.3. Sterilization methods” and “4.4. Investigation methods”.

Anyway, the actual structure of the manuscript is not properly made.

Regarding the investigation methods, the authors must pay attention to the methods like scanning electron microscopy that reveal the surface deterioration and the mechanical properties that was modified due to the sterilization methods.

I suggest to the authors to check some references who describe some evaluation methods like SEM .

Author Response

Reviewer 2

Answer

thanks for the suggestions and advice given to me to improve the manuscript

(The manuscript “The Effects of Sterilization Procedures on the Cutting Efficiency of Endodontic Instruments: Sistematic Review and network meta-analysis” must be revised seriously before publication. A lot of grammar mistakes are present and most of the figures from manuscript are unclear.Also, the authors have some serious problems with the terms related to the materials (e.g. line 235-237 “3.2 risk of bias The Risk of Bias”, Stainless stell) and they are advised to check carefully the manuscript before resubmission).

  • The manuscript was submitted to the mdpi English Editing Services, for the correction of grammatical and language errors, and changes were made for better clarity of the manuscript

The part “3. Results” must be correlated with the part “4. Discussion”. I suggest to use just 3.3. Metanalysis results instead of “3.3 metanalysis, 3.4 Network metanalysis, 4.3 meta-analysis results”.Also, in part 4, the discussion must be related to the aspects “4.1. Cutting efficiency”, “4.2.Materials”, “4.3. Sterilization methods” and “4.4. Investigation methods”.

Anyway, the actual structure of the manuscript is not properly made.

  • The structure of materials and methods and results has changed as suggested to me and also accepting the suggestions of reviewer 3

Regarding the investigation methods, the authors must pay attention to the methods like scanning electron microscopy that reveal the surface deterioration and the mechanical properties that was modified due to the sterilization methods.

I suggest to the authors to check some references who describe some evaluation methods like SEM .

  • Paragraphs concerning SEM and AFM have been added to the investigation methods as suggested
  • SEM analysis highlighted the presence of topographical alterations on the surface. These alterations were visible already after the first cycle in the autoclave and can represent points where microcracks are triggered within the structure of the lega and determine cyclic fatigue failure or torsional stress [60,61].
  • The alterations induced on the surfaces, analyzed by SEM and AFM, are also expressed on the cutting surface, detrimentally altering the cutting efficiency.
  • Surface analysis conducted by Inan et al., with the AFM reporting statistically significant data for all the instruments of the ProTaper series, noted the greatest effects on the ProTaper Finisching [62].
  • In 2012, Spagnuolo et al. confirmed that multiple sterilization cycles (autoclaving) changed the surface topography (examination conducted with AFM) and the chemical composition of conventional NiTi (F2 ProTaper) and TiN coated instruments (kit alpha) after five autoclave cycles [63].
  • Based on SEM analysis, Razavianet et al. reported an increase in roughness directly related to the number of sterilizations performed [64], while Nair et al. showed that, following sterilization, there is an increase in the surface roughness of the instruments due to an increase in the surface irregularities of the metal alloy, which could represent the cores from which the crack starts and the fracture of the tool occurs under cyclical fatigue [65].

Reviewer 3 Report

The manuscript  “The Effects of Sterilization Procedures on the Cutting Efficiency of Endodontic Instruments:. Sistematic Review and network meta-analysis” aimed to provide un update on the influence of autoclave cycle of sterilization on the cutting effect of endodontic instruments.

The subject is interesting despite of the fact that the analyzed papers were from 1983, 1989, 1996, 1998, 1999, 2002 and 2015 and, since then a lot of changes occurred in manufacturing techniques and materials of dental instruments.

Some issues need however to be addressed:

Title:

Please replace “Sistematic” with “Systematic” (row 2)

Abstract:  

Please replace „SCURA” with “SUCRA” (surface under the cumulative ranking). Please perform the above-mentioned changes in the entire manuscript (ex. Rows 357, 372, 511).

Please clarify the abstract: according to the author’s systematic review and meta-analysis what is the recommended protocol of sterilization (for endodontic instruments) and what is the life-span of stainless steel and Ni-Ti instruments for avoiding accidents during their use.

These aspects should be pointed-out in the discussions of meta-analysis results.

Some other aspects need to be discussed:

  • Heat sterilization – is not very accurate for autoclave, involving, beside heat, steam and moisture. Also, the instruments could be packed or not. This is an aspect to be discussed.

A deeper focus on the systematic review will be useful for the reader and will clarify some of the above-mentioned issues.

Some minor issues:

  • Please remove “a” from “includes a several phases” (row 52)
  • Please rephrase (clarify) rows 78-82
  • Please replace “magazine” with “Journal” (row 227)
  • Please replace “Stainless stell” with “stainless steel” in all manuscript (ex. Rows 490, 491, 497, Table 2).
  • Please improve clarity of figures 2, 3, 4, 6, 7
  • Please replace “Eeffects” with “Effects” (row 456)
  • Please change the structure of paragraph 4.1 I suggest in to include in 4.1.2 Effects of sterilization on Cutting efficiency, 4.1.2.1 Ni-Ti and 4.1.2.2 Stainless-steel

Author Response

Answer

thanks for the suggestions and advice given to me to improve the manuscript

  1. (Please replace “Sistematic” with “Systematic” (row 2): replaced
  2. Please replace „SCURA” with “SUCRA” (surface under the cumulative ranking). Please perform the above-mentioned changes in the entire manuscript (ex. Rows 357, 372, 511).: replaced
  3. Please clarify the abstract: according to the author’s systematic review and meta-analysis what is the recommended protocol of sterilization (for endodontic instruments) and what is the life-span of stainless steel and Ni-Ti instruments for avoiding accidents during their use.The alterations in the effects on cutting efficiency would seem to trigger after 5 cycles of sterilization by heat (autoclave).: added the following sentence in the abstract :in conclusion, the meta-analysis of the data indicates that the autoclave sterilization protocol must not be repeated more than 5 times, with the limitations of this review we can therefore establish that sterilization by autoclaving alone makes steel and Niti instruments less efficient after 5 cycles for a reduction in cutting efficiency
  4. Heat sterilization – is not very accurate for autoclave, involving, beside heat, steam and moisture. Also, the instruments could be packed or not. This is an aspect to be discussed: added the following sentence: Heat sterilization can be performed with 3 different methods:

steam sterilization (autoclaves): the dried and packaged instruments undergo a heating cycle (transmitted through the steam that penetrates the packaging) which can vary from 134 to 121 degrees Centigrade for a time that can vary from 5 to 30 minutes [36];

 Dry heat sterilization : the heat is not transmitted through the steam but through the air that can be forced to circulate at a temperature of 190 degrees for a period of time ranging from 5-15 minutes up to 2- 4hours dependent on the packaging method (metallic packages or packages out of plastic laminates that can withstand much heat) [37];

Sterilization by chemical steam (chemiclaves): sterilizing agent used and a chemical steam (alcohol and formaldehyde) suitably heated. The sterilization cycle is 20 minutes at 134 degrees [38,39].

  1. Please remove “a” from “includes a several phases” (row 52): removed
  2. Please rephrase (clarify) rows 78-82: Scientific studies therefore report conflicting opinions on the alterations affecting endodontic instruments subjected to sterilization procedures, this divergence of opinions and results could depend on the countless instruments and protocols used in endodontics as well as on the different sterilization and disinfection methods [1].
  3. Please replace “magazine” with “Journal” (row 227): Replaced
  4. Please replace “Stainless stell” with “stainless steel” in all manuscript (ex. Rows 490, 491, 497, Table 2).: Replaced
  5. Please improve clarity of figures 2, 3, 4, 6, The images have been increased in size and a better explanation of their meaning has been added to some
  6. Please replace “Eeffects” with “Effects” (row 456): Replaced
  7. Please change the structure of paragraph 4.1 I suggest in to include in 4.1.2 Effects of sterilization on Cutting efficiency, 4.1.2.1 Ni-Ti and 4.1.2.2 Stainless-stee .the structure of the manuscript and in particular of the materials and methods section and of the results has been modified in consideration of your suggestions and the advice given to me by reviewers 1 and 2
  8. the manuscript was corrected for English editing errors by English Editing Services

Round 2

Reviewer 3 Report

The manuscript is now considerably improved according to the reviewers' comments, and the concerns are properly addressed.

This manuscript is a resubmission of an earlier submission. The following is a list of the peer review reports and author responses from that submission.

Round 1

Reviewer 1 Report

The systematic review aimed to investigate the alterations induced by the disinfection and sterilization procedures on the cutting efficiency of endodontic instruments.

The authors presented an extensive qualitative analysis and quantitative analysis to fulfil this objective. Furthermore, the PRISMA Guidelines were followed throughout the text and therefore, the present work was presented in a structured manner. In addition, the work seems pertinent, updated, and may provide useful information for clinical application.

The quality analysis of the included studies was performed adequately, and the report of the results match the data extraction process described in the material and methods section.

Nevertheless, as a minor suggestion, this reviewer suggests that the authors disclose the possible limitations of the systematic review in the discussion section, as stated specifically by one of the items from PRISMA.

Author Response

Reviewer 1

The systematic review aimed to investigate the alterations induced by the disinfection and sterilization procedures on the cutting efficiency of endodontic instruments.

The authors presented an extensive qualitative analysis and quantitative analysis to fulfil this objective. Furthermore, the PRISMA Guidelines were followed throughout the text and therefore, the present work was presented in a structured manner. In addition, the work seems pertinent, updated, and may provide useful information for clinical application.

The quality analysis of the included studies was performed adequately, and the report of the results match the data extraction process described in the material and methods section.

Nevertheless, as a minor suggestion, this reviewer suggests that the authors disclose the possible limitations of the systematic review in the discussion section, as stated specifically by one of the items from PRISMA.

Answer

Thanks for the suggestions and comments made on the manuscript.

added the following sentence in the discussion

The limitations of this meta-analysis are the small number of included studies that present a low risk of bias, due to the different types of endodontic instruments investigated and the often different cutting efficiency detection methods.

Reviewer 2 Report

The manuscript “The Effects of Sterilization Procedures on the Cutting Efficiency of Endodontic Instruments. Sistematic Review.” looks interesting but many aspects needs improvements.

The manuscript didn’t bring more information than the previous published papers, like the paper published in International Journal of Dentistry, respectively Management of Instrument Sterilization Workflow in Endodontics: A Systematic Review and Meta-Analysis, Mario Dioguardi,1 Diego Sovereto,1 Gaetano Illuzzi,1 Enrica Laneve,1 Bruna Raddato,1 Claudia Arena,1 Vito Carlo Alberto Caponio,1 Giorgia Apollonia Caloro,2 Khrystyna Zhurakivska,1 Giuseppe Troiano,1 and Lorenzo Lo Muzio1,, Review Article | Open Access, Volume 2020 |Article ID 5824369 | https://doi.org/10.1155/2020/5824369 

Anyway, the authors could find below some comments in order to improve the manuscript before submission in other form.

Authors perform an interesting analysis, following some clear methods from statistic point of view. The interpretation of the results must be improved.

In Section 2.3.Screening methodology appear some mistakes. Try to reformulate more clearly this section.

In table 2, component “Instruments”, the authors could add all details about materials and the surface (some details appears like NiTi, PVD, stainless steel, but not for all instruments mentioned in this table).

Figure 3. 4 and 6 is not clear.

In section 4. Discussion, the authors made a long discussion about NiTi instruments. Please add some discussions about other metallic endodontic instruments.

The authors mention at line 208-209 about “Investigation methods to detect alterations and the main results of the included studies are also reported”. I suggest adding in “section 4. Discussion” more details about this (maybe a subsection 4.2.). Also, this aspect could appear in conclusion section.

Author Response

Reviewer 2

The manuscript “The Effects of Sterilization Procedures on the Cutting Efficiency of Endodontic Instruments. Sistematic Review.” looks interesting but many aspects needs improvements.

The manuscript didn’t bring more information than the previous published papers, like the paper published in International Journal of Dentistry, respectively Management of Instrument Sterilization Workflow in Endodontics: A Systematic Review and Meta-Analysis, Mario Dioguardi,1 Diego Sovereto,1 Gaetano Illuzzi,1 Enrica Laneve,1 Bruna Raddato,1 Claudia Arena,1 Vito Carlo Alberto Caponio,1 Giorgia Apollonia Caloro,2 Khrystyna Zhurakivska,1 Giuseppe Troiano,1 and Lorenzo Lo Muzio1,, Review Article | Open Access, Volume 2020 |Article ID 5824369 | https://doi.org/10.1155/2020/5824369

Answer

Thanks for the suggestions and comments made on the manuscript.

  • the previous review mainly dealt with the management of sterilization of endodontic instruments, and the best methods to obtain it by drafting a proposal for a protocol, and were treated in a generic way any alterations that sterilization procedures may induce in endodontic instruments

Anyway, the authors could find below some comments in order to improve the manuscript before submission in other form.

Authors perform an interesting analysis, following some clear methods from statistic point of view. The interpretation of the results must be improved.

In Section 2.3.Screening methodology appear some mistakes. Try to reformulate more clearly this section.

Answer

  • some sentences in section 2.3 have been changed and corrected as recommended

In table 2, component “Instruments”, the authors could add all details about materials and the surface (some details appears like NiTi, PVD, stainless steel, but not for all instruments mentioned in this table).

Answer

  • modified table 2 as required

Figure 3. 4 and 6 is not clear.

Answer

  • added a broader explanation below figures 3, 4, and 6, figure 6 has been revised and minimally improved in the graphical aspect.

In section 4. Discussion, the authors made a long discussion about NiTi instruments. Please add some discussions about other metallic endodontic instruments.

Answer

  • added as required by the following section :

4.2 Stainless stell

The effects of the autoclave on the mechanical properties of stainless stell instruments have been known for a long time Mitchel et al reported in 1982 a significant reduction in the resistance to torque and with a reduction of the deflection angle during torsional stress after 10 autoclave cycles [53], data partially confirmed by Haiknel et al. [54] while Hilt et al. reports data indicating that neither the number of sterilization cycles nor the type of autoclave unit affect the torque, hardness and microstructure of stainless steel in a statistically significant manner [55].

The studies on the cutting efficiency of stainless stell endodontic instruments present in the literature and included in this review are 3: Morrison et al. 1989  [27] reports no statistically significant difference for flexofile instruments from 1 to 15 sterilization cycles; Neal et al. 1983 [26]  describes a small but significant decrease in the cutting capacity of files (K-type 30 #) after 10 autoclaving cycles; Haikel et al. 1996 finds for the Unifile and Flexofile, a cutting efficiency reduction (range of 20 to 70%) from 5 to 10 autoclave cycles [22].

The authors mention at line 208-209 about “Investigation methods to detect alterations and the main results of the included studies are also reported”. I suggest adding in “section 4. Discussion” more details about this (maybe a subsection 4.2.). Also, this aspect could appear in conclusion section.

Answer

  • added as required by the following section : 1.1 Investigation methods

Haikel et al  [46]describe in one article different methods used for measuring the cutting effectiveness, the techniques used can be classified into two groups based on whether the tested instruments underwent a linear [47]or rotary movement along its long axis[46,48], moreover, the simulated samples can be in plexiglass, dentin[49] or bovine bone[50].

The first study on the cutting efficiency of endodontic instruments was conducted by Molven et al., dentin from sections of the extracted premolars was used as study sample and the instruments were used with mechanical and manual push and pull movements (linear motion) and the created sulcus surface was measured without checking the load on the instrument [49].

Previously to Molven et al, Fromme, and Riedel had qualitatively assessed the cutting efficiency of reamers, H files and pulling them through evaluations of images performed on tooth sections through the scanning electron microscope (SEM)[51].

Oliet was the first to introduce bovine dentine samples (with increased sample standardization) for the evaluation of cutting efficiency on reamers, using a mechanical rotary motion with torque control without irrigation and the depth of the grooves was measured with millimeter scale [50].

Villalobos et al. He investigated the cutting efficiency of the tip with rotary drilling movements on bovine bone maintained in humid conditions with measurement of efficiency through the measurement of time[48]. Instead Webber et al. has developed a system to measure the cutting efficiency in linear motion on blocks of bovine bone by testing different steel endodontic files with different section designs (square reamer, square file, triangular file, triangular reamer, Hedstroem file)[47].

In the studies included in the meta-analysis, the measuring devices involved measuring the depth of cut. Morrison measures the depth of cut on an apparatus (Quality Dental Products, Johnson City, TN) that compares sharpness of files when used in linear motion with irrigation and the samples used were blocks of phenolic resin [27] .

Neal also measures the depth of cut on methacrylate resin blocks with linear traction movement with irrigation, while Schafer et al. (2002) [21]  adopt, to measure cutting efficiency, the depth of cut on plastic cylinders generated by a mechanical rotary movement of the endodontic files [26].

Reviewer 3 Report

Page 2 L 53-55. This is very old  and out of used and shouldn't be address. Not even as an historical data. Will create confusion in the reader.

L 60-75. Please mention the type of apparatus and methodologies used in all these different studies and address it in your discussion.

your tables looks to blurry (as copy/paste) please be sure to fix this issue.

What is the manufacturer recommendation on regards the use of their rotary files? single use? Do they recommend to sterilize their files? 

Author Response

Reviewer 3

Page 2 L 53-55. This is very old  and out of used and shouldn't be address. Not even as an historical data. Will create confusion in the reader.

Answer

Thanks for the suggestions and comments made on the manuscript.

removed the sentence as required

L 60-75. Please mention the type of apparatus and methodologies used in all these different studies and address it in your discussion.

  • added as required by the following section : 1.1 Investigation methods

Haikel et al  [46]describe in one article different methods used for measuring the cutting effectiveness, the techniques used can be classified into two groups based on whether the tested instruments underwent a linear [47]or rotary movement along its long axis[46,48], moreover, the simulated samples can be in plexiglass, dentin[49] or bovine bone[50].

The first study on the cutting efficiency of endodontic instruments was conducted by Molven et al., dentin from sections of the extracted premolars was used as study sample and the instruments were used with mechanical and manual push and pull movements (linear motion) and the created sulcus surface was measured without checking the load on the instrument [49].

Previously to Molven et al, Fromme, and Riedel had qualitatively assessed the cutting efficiency of reamers, H files and pulling them through evaluations of images performed on tooth sections through the scanning electron microscope (SEM)[51].

Oliet was the first to introduce bovine dentine samples (with increased sample standardization) for the evaluation of cutting efficiency on reamers, using a mechanical rotary motion with torque control without irrigation and the depth of the grooves was measured with millimeter scale [50].

Villalobos et al. He investigated the cutting efficiency of the tip with rotary drilling movements on bovine bone maintained in humid conditions with measurement of efficiency through the measurement of time[48]. Instead Webber et al. has developed a system to measure the cutting efficiency in linear motion on blocks of bovine bone by testing different steel endodontic files with different section designs (square reamer, square file, triangular file, triangular reamer, Hedstroem file)[47].

In the studies included in the meta-analysis, the measuring devices involved measuring the depth of cut. Morrison measures the depth of cut on an apparatus (Quality Dental Products, Johnson City, TN) that compares sharpness of files when used in linear motion with irrigation and the samples used were blocks of phenolic resin [27] .

Neal also measures the depth of cut on methacrylate resin blocks with linear traction movement with irrigation, while Schafer et al. (2002) [21]  adopt, to measure cutting efficiency, the depth of cut on plastic cylinders generated by a mechanical rotary movement of the endodontic files [26].

your tables looks to blurry (as copy/paste) please be sure to fix this issue.

Answer

I have revised the format of the tables as required.

What is the manufacturer recommendation on regards the use of their rotary files? single use? Do they recommend to sterilize their files?

Answer

Most of the manufacturers supply rotating instruments as disposable in sterile blisters, it is advisable to clean the instruments anyway before use to eliminate any small metal residues before clinical use.
